# Designing and Developing a Population/Literature-Based Westernized Diet Index (WDI) and Its Relevance for Cardiometabolic Health

**DOI:** 10.3390/nu17142314

**Published:** 2025-07-14

**Authors:** Miguel Cifuentes, Zahra Hejazi, Farhad Vahid, Torsten Bohn

**Affiliations:** 1Nutrition and Health Research Group, Department of Precision Health, Luxembourg Institute of Health, 1445 Strassen, Luxembourg; miguel.cifuentes@lih.lu (M.C.); zahra.hejazi@lih.lu (Z.H.); 2Doctoral School in Science and Engineering, University of Luxembourg, 2, Avenue de l’Université, 4365 Esch-sur-Alzette, Luxembourg

**Keywords:** NOVA system, FFQ, secondary plant metabolites, gut microbiota, chronic diseases, Mediterranean diet

## Abstract

**Background/Objectives**: Recent research indicates a global transition from healthy and balanced diets to unhealthy Westernized dietary patterns (WDPs). This transition is linked to increased rates of non-communicable diseases (NCDs), e.g., obesity, type 2 diabetes, and cardiovascular diseases, often preceded by metabolic syndrome (MetS). Therefore, the objective of this study was to develop a diet quality index, termed Westernized Diet Index (WDI), to assess adherence to WDPs and its association with main cardiometabolic health issues, for which MetS and its components were chosen as representatives of NCDs. **Methods**: The development of the WDI was driven by a semi-systematic and comprehensive examination of the literature (*n* = 491 articles) that evaluated the influence of WDP components on health outcomes. The scoring algorithm involved multiple steps, assigning scores based on study design, sample size, and the direction of food effects on health outcomes. **Results**: The final developed index encompassed 30 food groups/items. It was revealed that soft drinks, processed foods, red meat, sodium, and hydrogenated fats had the most detrimental effects on health, significantly influencing the index’s coefficients. In contrast, dietary fiber, plant-based metabolites, vitamins, minerals, nuts/seeds, and fish had the most substantial beneficial impacts. **Conclusions**: The WDI aligns with the existing literature on the importance of specific food items and with other validated diet quality indices, e.g., the Dietary Inflammatory Index (DII) and Alternate Healthy Eating Index (AHEI). Thus, the WDI can provide evidence for clinicians and researchers in formulating evidence-based dietary guidelines as well as strategies for the prevention and treatment of diet-related health issues. However, further validation is proposed to verify the WDI’s capability across different contexts.

## 1. Introduction

Several factors, such as the successful control of infectious diseases and a shift in population lifestyle towards a more sedentary and Westernized pattern, are associated with a steep increase in non-communicable diseases (NCDs) in recent decades. NCDs are related to around 70% of global mortality, constituting a major socioeconomic burden [1,2,3,4]. In contrast to infectious diseases, NCDs are characterized by modifiable risk factors such as smoking, excessive sodium intake, alcohol consumption, a sedentary lifestyle, and, most notably, following an unhealthy dietary pattern [5]. Among NCDs, metabolic syndrome (MetS) is not only a prevalent risk factor but also a critical mediator in the development of cardiovascular diseases (CVDs), endocrine dysfunction, metabolic disorders, and other comorbidities (Appendix A) [2].

The worldwide prevalence of MetS has increased in close correlation with the prevalence of overweight, obesity, and type 2 diabetes (T2D) [6]. Since 1990, the global prevalence of adult overweight and obesity has more than doubled, with 43% and 16% of adults being overweight (body mass index (BMI) between 25 and 30 kg/m^2^) and having obesity (BMI > 30 kg/m^2^), respectively, in 2022 [7,8]. Accordingly, MetS affects over 25% of adults around the globe [9], as well as nearly 40% of older people in their sixth decade of life [10]. Evidence suggests that MetS incidence and prevalence are mainly associated with lifestyle pattern alterations, such as poor dietary habits (e.g., following a Westernized diet (WD)), built environment (e.g., urbanization), reduced physical activity (e.g., sedentary lifestyle), and high exposure to pollutants [4,11,12,13].

Dietary patterns, as modifiable risk factors [2], are one of the main contributors to chronic diseases and play a significant role in the incidence/prevalence of NCDs, particularly MetS [14]. However, recent research indicates a global shift in eating trends from traditional healthy diets to a WD. This shift in eating patterns has become especially pronounced in developing countries [15,16].

Updated data regarding MetS prevalence and food consumption worldwide clearly illustrate the association between the WD and MetS [17,18]. For instance, the prevalence of MetS in Western Europe and North America is 22% and 34%, respectively. These numbers contrast when compared to 15% in Africa and 14% in Southeast Asia [17,19]. Mirroring MetS prevalence in the global dietary database map [18] highlights the impact of the WD trend on MetS. Some examples of these regional differences are reflected by the intake of cholesterol, processed meat, and sugar-sweetened beverages (SSBs) as markers of animal products, processed food intake, and simple sugar consumption, respectively.

Comparing cholesterol intake in the USA and India, the mean daily intake was 289 mg and 132 mg, respectively, correlating with higher MetS rates in the USA (34%) vs. India (25%) [17,18,19,20]. Similarly, processed meat consumption in the US is approximately nine times higher than that in Bangladesh (37 g/day vs. 4 g/day, respectively), while MetS prevalence in Bangladesh is 16% (vs. 34% in the USA) [18,21]. Likewise, SSBs consumption in Mexico has been reported as 375 ml/day, whereas in Nepal, this was 28 mL/day, and the prevalence of MetS in these two countries is 44% [22] and 15% [23], respectively [18]. These examples indirectly show the association between some of the WD components and the prevalence of MetS. However, the study of individual food item intake may not fully represent Westernized dietary patterns, as the intake of healthy items may compensate for the intake of unhealthy food items. Thus, a comprehensive diet evaluation may be a more effective strategy to study such correlations. To obtain a more comprehensive overview of the entire diet, dietary indices may be the preferred method. These indices provide a more holistic assessment of eating patterns, considering multiple food groups and nutrients, which helps to better understand their impact on health outcomes.

In recent decades, several dietary indices, such as the Dietary Inflammatory Index (DII) [24], the Healthy Eating Index (HEI), the Alternative Healthy Eating Index (AHEI) [25], and the Dietary Antioxidant Index (DAI) [26,27], have been developed and validated in several studies. They are considered practical tools for assessing the link between diet quality and various health outcomes, including MetS and NCDs [28]. However, there is a noticeable gap in the literature when it comes to the WD, which is a relatively recent trend in dietary patterns. Thus far, dietary indices have struggled to accurately capture the composition of the WD and have tended to rely on a broader definition, often overlooking the specific dietary components that define the Westernized pattern. Therefore, due to the high prevalence of NCDs, as well as an increase in adherence to the WD in different populations, the objective of this study was to design and develop a population/literature-based Westernized Diet Index (WDI). We hypothesized that the WDI would be able to comprehensively assess people’s adherence to the WD, evaluate the quality of diets, and establish associations between diets and a wide variety of diseases and complications, including CVDs, T2D, non-alcoholic fatty liver disease, overweight/obesity, MetS, and cancers, among others.

## 2. Materials and Methods

### 2.1. Overview

The WDI was calculated based on an extensive examination of the literature relating the effects of individual WD components on MetS and its components, including high blood pressure (BP), high fasting blood sugar (FBS), high triglycerides (TG), high waist circumference (WC), and low high-density lipoprotein cholesterol (HDL-c).

### 2.2. Literature Review and Selection

We conducted a review of peer-reviewed scholarly publications published between 2001 and March 2024. The following sections describe the steps involved in the practical definition of keywords and provide a detailed account of the WDI development and scoring algorithm methodology.

Eligibility criteria for this study included primary research evidence from desirable study designs according to evidence-based public health, such as clinical trials, cohort studies, cross-sectional (case/no case design) studies, and descriptive designs [29]. However, review articles were excluded in order to prevent information redundancy; additionally, case series and case reports were omitted due to their lower rank in the evidence-based public health hierarchy, considering that such studies could result in less robust and generalized conclusions [29]. Comments, congress proceedings, editorials, book chapters, and retracted publications were also excluded. Furthermore, the study had to evaluate the effect of a WD component on MetS and/or its components.

Selected studies used different methods to evaluate dietary intakes (e.g., food frequency questionnaires (FFQ), 24-h recall, 3-day records, etc.) to explore the effect of food items or groups on MetS and/or its components. Only articles in English were included. Animal and cellular model studies were excluded due to the significant difference between their MetS definition compared to humans and differences in nutrient metabolism. Other aspects, such as age, sex, quantity of food, evaluation methods, statistical approaches, geographical area, location, and study limitations, were not considered inclusion/exclusion criteria (Table 1).

In this study, the WD and its components were determined using a two-step approach. First, a semi-systematic approach was used to gather relevant information. Second, expert opinions were incorporated through discussions with team members involved in the project, ensuring the inclusion of expert insights to refine and validate findings. These expert judgments were used to include any further components that were overlooked in the literature. Briefly, the top 15% (n = 144) of the best-matching articles following the search for “Western Diet” in PubMed were reviewed. This aimed to consolidate the definition of the components of the WD. The expert list of WD components was combined with the literature results, and descriptive statistics were used to summarize the combined components. The final list of WD components can be found in Appendix A.

There was no preference for any MetS definition (Appendix A) while developing the index, given their similarity. Hence, all definitions were included in the development of the index. MetS was introduced as a MeSH term in 2001 in PubMed; therefore, the time limitation of this study was from 2001 onward to keep the standardized and complete definition of MetS.

### 2.3. Search Strategy

The PubMed search engine was used to retrieve peer-reviewed articles published in English that met the established criteria (Table 1). The search strategy consisted of (i) the combination of MetS and WD components as the main pillars of the study, linked with the AND function from Boolean operators; (ii) exclusion of ineligible publication types (see previous section) using the NOT function; and (iii) other filters (exclude preprints, only on human studies, and in the English language) (Appendix A). All authors checked and validated the search strategy during the design phase. The results of this search were then transferred to the free, web-based tool CADIMA [31] for screening and data extraction phases (Figure 1). Consistency checks were carried out prior to the beginning of each screening phase to align the criteria.

The full texts of articles matching the criteria (n = 565) were evaluated. Of these, 61 were excluded (Figure 1). In all screening phases, any inconsistencies and/or conflicting studies were revised by a third reviewer.

In the data extraction phase, 13 articles were excluded due to the incompatible data reporting methods. Finally, 10 articles were excluded at the scoring stage, as the food items/group could not fit into the defined food groups, i.e., using too general terms such as healthy diet, complex diets, and composite dishes (Figure 1).

### 2.4. Data Extraction

First, consistency checks were conducted by the two reviewers, followed by modifications to the extraction method and the precise information retained from the articles. For each of the 504 articles (Appendix A), information was gathered about the study design (StD), population, age range, sex, sample size, intake evaluation methods, amount of food component consumed, length of intervention (if applicable), statistical approach, limitations, conclusions, definition of MetS, and the effect direction of the food component on MetS and/or its components.

Data were further classified into positive, negative, or no-effect groups according to the associations found in each study. For example, if the food component had a positive, i.e., ameliorating effect, on MetS and/or its components, it would be placed in the positive effect group. By contrast, if the food component had a detrimental effect on MetS and/or its components, it was placed in the negative effect group. Lastly, if there were no significant effects on the MetS/its components, it was placed in a third group, labelled “non-significant”.

### 2.5. Scoring Algorithm

The scoring methodology involved several key steps, including data coding, categorization, and calculations. Following the completion of the data extraction phase, the dataset underwent a thorough review by both reviewers for further cleaning and synchronization.

Included studies in the dataset were classified into eight distinct groups based on StD and evidence-based public health and medicine [29]: (i–iii) clinical trials (comprising randomized and blind trials, randomized or blinded trials, and non-randomized and non-blinded trials), (iv) cohort studies, (v) case-control designs, and (vi–viii) cross-sectional studies (including nested designs, case-no case designs, and descriptive designs). In addition, each study within a StD was allocated into tertiles (T1, T2, T3), according to the sample size (i.e., number of study participants). As an example, in descriptive designs, the first, second, and third tertiles correspond to sample sizes of up to 290, 637, and 25,506 participants, respectively.

The original dataset was further labeled with food groups/items and a corresponding matrix consisting of labeled StD, labeled tertile for each StD, and labeled food groups derived from the completed dataset. The coded dataset was utilized in subsequent steps to assign scores to each food group. For each food group, a table was created showing the sum of the articles with a certain StD, tertile, and the food group/item effect direction (positive, negative, and non-significant; see Section 2.4 on selected health outcomes.

The scoring matrix for each food group/items/components consisted of rows representing the StDs and tertiles and columns representing the effect direction of the food group on MetS and each of the components. Each entry in this table was assigned a predefined score based on the strength of the StD, tertiles of the number of subjects according to the StD, and the effect direction (positive, negative, or non-significant) on health outcomes. The scores for StD were based on evidence-based public health [29] and previously published studies [24]. The studies in T1, T2, and T3 in each StD were given 1, 2, and 3 points, respectively (Table 2).

### 2.6. MetS/Its Components and Score Calculation

The scores for MetS and its components in each StD group were calculated in three main steps:

In the first step, the StD score (see Table 2) and the third tertile score were summed (values in T3 were used in this step to ensure that MetS and its components had the highest score) in each StD category. This was conducted to highlight the influence of the health outcome on the index. In the second step, the overall score for MetS and its components was calculated by multiplying the score previously calculated by 3 (fixed factor used to increase the weight of MetS and its components when compared to the weight of the sum of the StD and sample size tertile score). Third, to obtain the MetS score (for positive, negative, or non-significant effects) when only MetS was reported (e.g., no MetS components), the factored score in the second step was divided by two. Furthermore, the score for each component of MetS reported was calculated by dividing the MetS score by five (as MetS is comprised of five components according to several definitions) (Table 3 is an example of scoring in cohort studies).

The next step was to obtain the reference scoring matrix, which would later be used to calculate each food item/group score. For this matter, the score of the previously calculated MetS or its components score was multiplied by the sum of StD and tertile score (Table 4). This complex scoring algorithm was designed to provide a comprehensive evaluation of the impact of various food groups/items on MetS and its components, accounting for differences in the StD and sample sizes.

In the last step, each food group matrix was multiplied by the overall scoring matrix. Then, the sum of the positive columns, negative columns, and non-significant columns was calculated separately.

To reach the coefficient for each food group, the sum of the positive and negative values was divided by the absolute sum of the three values (Equation (1)). Moreover, Figure 2 provides a summary of the WDI scoring steps, outlining the methodology used to calculate the index. It visually represents the key stages involved in assigning scores based on StDs, sample sizes, and the effect of dietary components on health outcomes. Table 5 provides an example of the score calculation for the “vitamins and minerals” group. The same approach was consistently applied to all 30 items in the index.

Calculation of each food group/item score:(1)$$\begin{array}{l} Food\;groups/items/componentsscore=\\ \frac{Sum\;of\;positive\;effect\;scores+Sum\;of\;negative\;effect\;scores}{Absolute\;sum\;of\;the\;columns\;(Sum\;of\;positive\;effect\;scores+\left|Sum\;of\;negative\;effect\;scores\right|+nonsignificant\;effect\;scores)} \end{array}$$


## 3. Results

The WDI was developed using a diverse range of age groups, with adults (18–65 years) being the most prevalent age group (76%) studied in the selected publications. Additionally, the gender composition was mixed, with 87% of publications including both male and female participants (Appendix A). Moreover, the majority of articles used the ATP-III definition, while 33% and 6% used the IDF and AHA definitions, respectively. This variety was chosen to add validity and applicability to the WDI across different clinical definitions, ensuring its reliability across different genders and age groups.

The primary outcome of this study was a comprehensive coefficient of the WDI per food group/item, i.e., a multicomponent score that captures various aspects of the diet (Table 6), with a higher absolute coefficient indicating a more substantial health impact (either positive or negative). Figure 3 provides a visual representation of the most prominent WDI components, considering both the number of supporting articles and the final coefficient assigned to each food group/item.

According to the final scoring algorithm and results (Table 6), soft drinks (−0.37), processed foods (−0.31), red meat (−0.30), sodium (−0.28), and hydrogenated fat (−0.27) were the top five food groups/items with the highest negative WDI coefficients, indicating an adverse effect on MetS and/or its components. On the other hand, dietary fiber (0.40), secondary plant-based metabolites (0.34), vitamins and minerals (0.28), nuts and seeds (0.27), and fish (0.25) had the highest WDI coefficients, indicating an ameliorating effect on health outcomes. The calculated coefficients will later be used to validate the index and to develop a calculation algorithm.

## 4. Discussion

Following a population/literature-based study, we developed a nutritional assessment tool to evaluate adherence to unhealthy Westernized dietary patterns. The developed index incorporates a number of relevant food groups and components that have been associated with westernized patterns and cardiometabolic health outcomes. The derived coefficients emphasize the positive or negative strength of the food component in the overall score with a relevant health outcome, i.e., MetS. By incorporating specific numerical scores for individual food components, the WDI provides a detailed assessment, capturing subtle variations in dietary patterns and offering a more comprehensive and precise evaluation of diet-related health risks. To our knowledge, this is the first dietary index to prioritize WD components and systematically assess their impact on health outcomes, particularly MetS.

Based on our scoring algorithm, the food groups/items with the most detrimental effects on the outcomes/endpoints (MetS and components) were soft drinks, followed by processed foods and red meat. On the other hand, dietary fiber, the sum of secondary plant metabolites, and the sum of vitamins/minerals as protective food groups showed the most positive coefficients. The coefficients are well aligned with the existing literature on MetS risk factors [32,33] and other developed and validated diet quality evaluation indices, e.g., DII, AHEI, and DAI, among others [26,28,34].

### 4.1. Negative Coefficients of the WDI

Among the most detrimental components of the WD, sugar-sweetened beverages (SSBs) ranked highest in our index, indicating their very strong negative impact on health outcomes. SSBs can contribute to poor health outcomes such as body weight gain, hyperinsulinemia induced by fast absorption of glucose [35], potentially inducing hypertriglyceridemia and disturbed hepatic metabolism due to excess fructose intake [35,36]. Furthermore, SSBs are considered the highest source of added sugar and the largest source of daily energy intake in the US diet [37], with intakes of three to four cans (>1 L) of soda per day. This tendency toward soft drink consumption can be considered a hallmark of the WD, and it is reported in many countries worldwide [38]. While SSBs are often discussed in the context of unhealthy dietary patterns, they are not explicitly included as individual components in some of the most widely used indices, as is the case with the MDS or AHEI [28]. However, in some other indices, such as the DII, SSBs may not be explicitly listed, but foods that promote inflammation (such as those rich in sugar) would influence the score within the broader category of sugary foods and drinks [39]. Only a few indices, such as the Mediterranean-Style Dietary Pattern Score (MSDP), consider SSB intake in their calculation. In this case, the effect of SSBs on the overall score was very pronounced [40], in a similar manner to our WDI. The inclusion of SSBs in our index highlights the importance of addressing this component when evaluating the overall effects of diet, particularly in relation to the WD.

The second-lowest-ranked food group in the WDI was processed and ultra-processed foods (UPFs), reflecting their significant negative impact on the overall score and highlighting their increasing recognition in recent years as a dietary component with major implications for various health outcomes. UPFs are characterized by high energy density, being rich in saturated fats, refined starches, free sugars, and salt, while low in dietary fiber, protein, and micronutrients [41]. In the US and different European countries, more than 60% of dietary energy originates from UPFs [42,43,44]. Furthermore, there is strong evidence showing the adverse effects of processed foods (especially UPFs) on health. Higher intake of UPFs was associated with a significantly increased risk of MetS and other markers of cardiometabolic health in a meta-analysis [45]. Considering the worldwide increase in processed food consumption, there is an essential need to investigate more in-depth aspects of their consumption. UPFs have nevertheless been neglected by most dietary indices so far [46]. Although some indices have tried to capture this component in recent years, one such index is the cardiovascular health index [47]. In this regard, the WDI not only introduces UPFs into dietary assessment indices but also showcases their relevance as a primary driver of adverse health outcomes.

The third negative component was red meat, with the percentage of calories derived from this food group in developed countries generally being higher than in developing countries [48]. The global average of meat consumption per capita increased by 75% between 1961 and 2011 [49]. However, meat consumption trends in developing countries predict a 2.9% growth in red meat (beef) consumption annually [48,50], indicating a transition to a more WD. Red meat consumption has been related to major chronic diseases, including CVDs and various types of cancer [51,52]. A meta-analysis of 43 observational studies in CVDs and 27 observational studies for diabetes showed that each 100 g/day increment in unprocessed red meat consumption was associated with a higher risk of chronic heart diseases, CVDs, and T2D [52]. Although some studies have found a positive association between red meat intake and health outcomes [53], our results favor an adverse effect of red meat intake. Recommendations to reduce red meat consumption are incorporated in some dietary indices, reflecting growing concerns about the health impact of excessive intake. A prominent example is the validated Mediterranean Diet Score (MDS) [28,54], where high red meat consumption is penalized. The MDS emphasizes plant-based foods, healthy fats, and moderate protein intake, offering a balanced dietary approach that reduces reliance on red meat [28,54]. However, the MDS is calculated based solely on food groups; moreover, overlooking quantitative scoring in this index may limit its ability to assess diet quality comprehensively [28]. The lack of detailed measurement makes it challenging to capture the relative impact of specific nutrients, reducing the index’s sensitivity to variations in dietary patterns and their effects on health outcomes [54]. A more nuanced, quantitative approach, as proposed in this WDI, could enhance the accuracy and effectiveness of the MDS in evaluating overall diet quality.

### 4.2. Positive Coefficients of the WDI

Dietary fiber showed the highest positive coefficient, reflecting its significant beneficial impact on health outcomes. This aligns with the literature, which often characterizes low intake of dietary fiber sources (i.e., fruits, vegetables, whole grains) as another hallmark of the WD [55,56,57]. For example, the mean dietary fiber intake in the US and France was reported at 16.5 g/d and 19 g/d, respectively, compared to 33.3 g/d in Uganda [58]. Furthermore, dietary fiber intake has been considered part of healthy and traditional diets such as the Mediterranean diet. In this regard, many indices emphasize the intake of dietary fiber as a key component. The DII, MDS, and AHEI, among others, are prime examples of indices that promote higher fiber intake for better health outcomes [54]. For instance, when comparing food components in the WDI coefficient table to the MSDPS [59] and AHEI [25], dietary fiber consistently ranks among the highest components in the three indices. In the case of the DII, dietary fiber had the second-highest anti-inflammatory position in the list of coefficients [24].

Secondary plant-based metabolites were ranked as the next highest positive coefficient component in the developed WDI. These include a wide range of different non-nutrient compounds, such as terpenoids, phenolic compounds, alkaloids, and sulfur-containing ones [58]. These metabolites are, to a large degree, considered natural antioxidants [60]. However, their metabolic activity reaches far beyond potential direct antioxidant effects and also includes transcriptional regulatory activity, for example, through interactions with NF-kB [61]. The importance of secondary plant metabolites is underscored by a meta-analysis of nine cross-sectional studies that assessed the impact of phytochemicals on cardiometabolic risks using the dietary phytochemical index (DPI) [62]. The analysis demonstrated a significant reduction in the risk of overweight/obesity with higher phytochemical intake [62]. The significance of secondary plant metabolites extends beyond their inclusion in the DPI; other indices, such as the DII, also incorporate various metabolites due to their well-documented anti-inflammatory properties [54].

Vitamins and minerals ranked as the third most positively associated component of the current index. A variety of factors, including the high consumption of processed foods and the low intake of fruits, vegetables, and whole grains, contribute to the failure of a significant portion of populations (e.g., in North America) to meet the recommended dietary allowances for vitamins and minerals [63]. As such, there is evidence that the status of Zn, folate, and vitamins C, D, E, and K may be compromised by a typical WD [64,65,66]. A low status of specific vitamins has been associated with cardiometabolic risk factors. Based on three large epidemiological cohorts, higher vitamin E intake from the background diet or supplementation was associated with a significant decrease in CVDs [64,66,67]. Although the importance of this group is well established, some commonly used indices fail to address vitamins and minerals in their assessment of diet quality adequately. For example, the MDS does not consider minerals or vitamins in its calculation. Other indices, such as DII or DAI, only include some (e.g., Fe, Vitamin A, Vitamin D, Zn), but they fail to capture total mineral or vitamin intake [54]. In our case, by providing a single group for these elements, we enabled a better quantification than that of other indices, where only specific items are considered, even though at the risk of summing up many different compounds.

### 4.3. The WDI Compared to Other Indices

The present index includes a broad range of food groups relevant to diet quality. Many of the food groups present within the WDI, such as fruits, vegetables, whole grains, nuts, and legumes, are common to many already developed and validated indices (e.g., MDS, AHEI) [54,59]. The similarities with other indices are not only found in food groups but also in nutrients and non-nutrient groups, such as secondary plant metabolites or minerals and vitamins (e.g., DII) [24]. However, some items are novel to the WDI. Therefore, it is anticipated to demonstrate stronger associations with health outcomes [68]. The most notable novel aspect of the WDI in this regard is the inclusion of processed foods, SSBs, and dietary supplements as independent components of the index. These considerations contrast with other indices. For example, the MSDPS [59], designed to assess the conformity of an individual to a traditional Mediterranean-style diet, was composed of only 13 components, not including processed foods, sodium, and total energy [69]; however, all are considered in the present WDI. This highlights the broad scope that the WDI provides in relation to the WD. In addition, the WDI offers users flexibility by providing a comprehensive list of food components used in each food group (Appendix A). An additional advantage of the WDI over other indices, such as the DQI-I with strong links to metabolic markers, is that those require a specific questionnaire for score calculation, while the WDI generates scores from any available dietary intake data, such as FFQs, 24-h recalls, or food records.

The inclusion of food groups in dietary assessment may provide an additional benefit. In a study comparing seven diet quality indices, it was shown that food-group-based indices such as MDS were more strongly related to health outcomes vs. rather nutrient-based ones (e.g., DII), perhaps due to the additional variability caused by translating food groups to nutrient intakes [33]. The WDI, as a nutrient/non-nutrient/food group-based index, covers various aspects of the diet and is probably able to cover the diet in its entirety quite comprehensively.

With respect to health outcomes, many frequently employed indices, such as the DII [24], DASH diet indices, DAI [26], and MDS [54], target specific aspects of diet or outcome. For example, the DII only focuses on the inflammatory aspect, the DAI only on the antioxidant properties of diet [54]. Given the broad target that the WDI was designed for, we have included a wider range of cardiometabolic health outcomes. Moreover, the WDI is able to capture not only adherence to WD but also to traditional patterns, as the components of our score have either a positive or a negative value. Meanwhile, most indices only have one-directional measurements, with undesired values not adding to the total score. For example, low adherence to MDS will have a score close to zero [28], although this does not necessarily indicate an unhealthy diet [54]. With the WDI, non-adherence to WD is rewarded with positive values that will push the final score towards a more traditional/healthy diet. Therefore, the WDI assesses, at the same time, traditional/healthy diets as well as adherence to Western/unhealthy patterns. This feature clearly differentiates the WDI from other broadly used indices, with the exception of the DII, and will likely boost its power in detecting associations between diet and cardiometabolic outcomes.

The comparative analysis of the WDI against other indices does not only feature the inclusion of novel components for assessing dietary indices, but also the mixture of food groups, nutrients, and non-nutrients. The latter feature has proven to be a solid approach in designing strong dietary indices that correlate with different endpoints (i.e., DII) [24]. Therefore, even though final validation is pending, the potential of the present index arises as a tool to guide dietary advice in public health. 

The developed WDI has several strengths, especially the wide range of studies included, considering the different study designs, sample size, populations, both genders, several cardiometabolic relevant endpoints, and considering a number of food items and groups, including some that are disregarded in other indices, such as SSBs and UPFs. The WDI was designed using a wide range of data comprising populations from all age groups, and thus it is expected to be applicable across those groups, including children and adolescents as well as adults and older people. Therefore, the WDI addresses the dietary risks leading to MetS and other related NCDs, while also providing a valuable tool for assessing dietary patterns. Another key advantage of the dietary index developed in this study is the continuous scoring approach, which allows a more precise and sensitive assessment of one’s diet. The continuous method can capture the full range of dietary intake, enabling the researcher or clinician to detect also minor variations in dietary patterns. Moreover, by avoiding predefined cut-offs, the continuous approach better reflects dietary behaviors and their changes over time. Since the WDI has been developed to be used in multivariate analysis, it holds no limitation in those studies in which further confounders may need to be controlled for and that may be part of westernized lifestyle patterns (i.e., socioeconomic status and physical activity).

There are some limitations as well. Although the WDI can capture the effect of a broad spectrum of food items on metabolic health, it may not completely embrace the complexity of diets, such as details about sub-classes of health-relevant food constituents, e.g., types of dietary fiber. It also does not consider the time of eating, i.e., Chrono-nutrition-related aspects. Likewise, the index does not include other relevant lifestyle factors such as physical activity, socioeconomic status, genetic variations, and other predispositions, although many other indices do not consider these either. Additionally, the reported data’s reliance on self-reported data such as FFQs or 24 h recall in selected studies may produce reporting biases. Nonetheless, this is a limitation common to all dietary indices and not to the WDI in particular, caused by the lack of adequate biomarkers of dietary intake [70]. Furthermore, the main limitation of the index, a validation in a real-life population, remains to be carried out. Without this step, it is not possible at present to know the robustness of the index and its association with cardiometabolic diseases. It will, therefore, remain a challenge for the validation step to confirm the sensitivity of the index and whether other weighing approaches may impact its power as a public health tool.

## 5. Conclusions

The developed WDI represents a valuable and robust, literature-based nutritional assessment tool for evaluating adherence to the WD, a dietary pattern that has become increasingly prevalent across diverse populations. The WDI’s utility extends beyond merely tracking dietary habits; it can play a crucial role in elucidating the association between the WD and the incidence and prevalence of NCDs. By quantifying the risks associated with the consumption of specific dietary components characteristic of the WD and a main, diet-related health outcome, i.e., MetS, the WDI offers insights that can inform targeted public health interventions. This capability is particularly significant in the context of rising NCD rates globally, where dietary factors are a primary modifiable risk factor. Moreover, the WDI is believed to serve as a robust tool for clinicians and researchers to more comprehensively investigate the intricate associations between dietary patterns and a wide range of health-related outcomes. To verify the index’s capability across different contexts, further investigation and validation are warranted.

## Figures and Tables

**Figure 1 nutrients-17-02314-f001:**
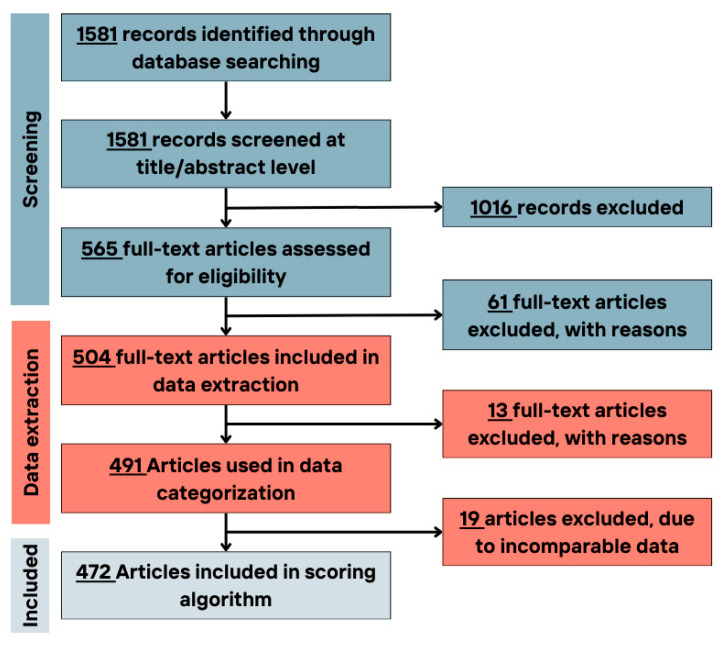
Flowchart of the number of included/excluded articles in different study phases.

**Figure 2 nutrients-17-02314-f002:**
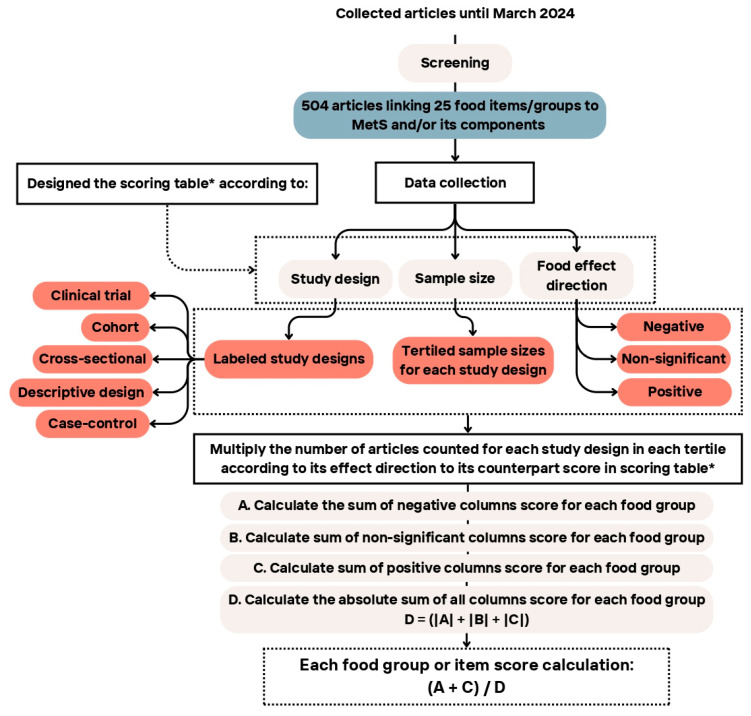
Study flow diagram demonstrating different steps of the Westernized diet index (WDI) development. * Details on the allocated points are given in Table 2, Table 3 and Table 4.

**Figure 3 nutrients-17-02314-f003:**
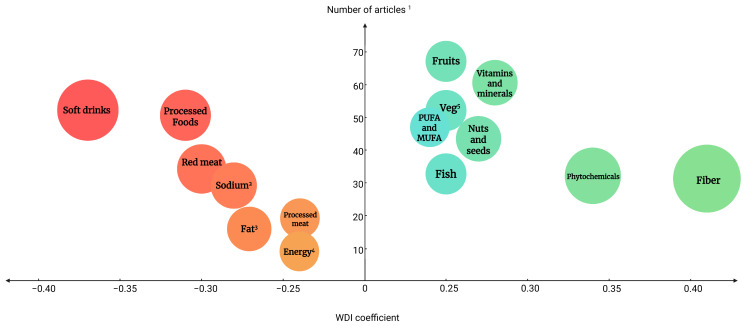
Balloon chart of the articles describing the investigated components vs. the strength of the coefficient. The size and color of the circles represent the strength of the coefficient, with shades of green indicating higher coefficients and shades of red indicating lower coefficients. Components with a WDI greater than 0.24 or less than −0.24 are shown in the graph. ^1^ Each reviewed article could contain more than one food component, ^2^ Sodium intake, ^3^ Hydrogenated fat, ^4^ Energy intake, ^5^ Vegetables, PUFA and MUFA, polyunsaturated fatty acids and mono-unsaturated fatty acids.

**Table 1 nutrients-17-02314-t001:** Summary of eligibility criteria and inclusion and exclusion criteria used in the screening phase of developing the WDI.

Aspect	Inclusion	Exclusion
**Study design**	Original peer-reviewed research papers	Case reports, reviews, systematic reviews, meta-analyses, comments, congress reports, editorials, book chapters, retracted publication
**MetS**	Containing MetS and/or its components in the study	No MetS and/or its components
**WD components**	Containing at least one of the WD components	No food components
**Evaluation**	Link between WD components and MetS and/or its components.	Studies not containing at least a WD component and its effect on MetS and/or its components.
**Population**	All populations, all ages, infants (≤12 months), toddlers (>12–36 months), children (>36 months–10 years), adolescents (>10–18 years), adults (>18–65 years), elderly (>65–75 years), very elderly (>75 years) [30]	N/A
**Region**	The whole world	N/A
**Time**	2001–Mar 2024	N/A
**Language**	English (abstract and whole text)	Non-English (abstract and whole text)
**Species**	Human studies (both genders)	Animal studies, cellular models, and in vitro studies

**Table 2 nutrients-17-02314-t002:** Study designs and tertile scores used to calculate the health outcomes score and overall score.

Study Designs (StDs)		StDs Coding	StD Score	StD + T1 Score	StD + T2 Score	StD + T3 Score ^^^
Clinical trials	Randomized, blinded clinical trials	I	10	11	12	13
	Randomized or blinded clinical trials	II	9	10	11	12
	Non-randomized and non-blinded clinical trials	III	8	9	10	11
Cohort		IV	7	8	9	10
Case-control		V	6	7	8	9
Nested cross-sectional		VI	5	6	7	8
Case-no case cross-sectional design		VII	4	5	6	7
Descriptive designs		VIII	3	4	5	6

T1: tertile 1 (1 point), T2: tertile 2 (2 points), T3: tertile 3 (3 points) of sample size in each StD. ^^^ The scores in these columns are later used to calculate MetS and its component scores.

**Table 3 nutrients-17-02314-t003:** Undertaken steps to calculate MetS and its component scores in cohort studies.

Step	Explanation	Calculation Detail		Score
1	Assigning the StD and tertile score to the cohort study ^^^	StD score	7 points	7 + 3 = 10 points
		Tertile score	3 points	
2	Increasing the effect of health outcomes in the final score (compared to the StD and tertile score)	StD and tertile score	10 points	10 × 3 = 30 points
		Factoring fixed number *	3 points	
3.1	Allocating scores to MetS (if reported)	Half of the calculated score in Step 2		30 ÷ 2 = 15 points
3.2	Allocating scores to each of the MetS components (if reported)	One-fifth of the calculated score from step 3.1		15 ÷ 5 = 3 points

^^^ The score for StDs and tertiles mentioned in Table 2, * a fixed number used to increase the effect of MetS and its components score higher than the sum of StD and sample size tertile score.

**Table 4 nutrients-17-02314-t004:** Example of scoring table for cohort studies reporting MetS positive, FBG negative, and DBP non-significant scores.

StD	Tertiles	MetS Positive ^^^	FBG Negative *	DBP Non-Significant §
Cohort (7 points)	T1 (1 point)	(7 + 1) × 15 = 120	−(7 + 1) × 3 = −24	24 ÷ 2 = 12
Cohort (7 points)	T2 (2 points)	(7 + 2) × 15 = 135	−(7 + 2) × 3 = −27	27 ÷ 2 = 13.5
Cohort (7 points)	T3 (3 points)	(7 + 3) × 15 = 150	−(7 + 3) × 3 = −30	30 ÷ 2 = 15

StD: Study design; T: tertile; FBG: fasting blood glucose; DBP: diastolic blood pressure. ^^^ enhanced MetS status by the food component, * deteriorated FBG status by the food component, § DBP non-significantly affected by the food component, systolic and diastolic blood pressure scores were calculated as half of the other components, as some of the studies reported one of them or separately. The values for the non-significant category are always positive.

**Table 5 nutrients-17-02314-t005:** The scoring algorithm is exemplified for one food group (vitamins and minerals). Numbers represent the number of articles found with specific StD and tertile, which had the same effect on different health outcomes. Numbers in brackets represent the total score of the articles in that category, calculated by multiplying the number of articles in each category by the overall score allocated to it. The calculation of the final score for the selected food group is also shown.

Study Designs ^a^	Tertiles ^b^	MetS ^c^ Positive	MetS Inverse	MetS Non-Significant	BG ^d^ Positive	BG Inverse	BG Non-Significant	HDL ^e^ Positive	HDL Inverse	HDL Non-Significant	TG ^f^ Positive	TG Inverse	TG Non-Significant	WC ^g^ Positive	WC Inverse	WC Non-Significant	DBP ^h^ Positive	DBP Inverse	DBP Non-Significant	SBP ^i^ Positive	SBP Inverse	SBP Non-Significant
**I**	T1	1 ^j^ (214.5) ^k^	-	-	-	1 (−42.9)	1 (42.9)	-	-	2 (85.8)	1 (42.9)	-	1 (42.9)	-	-	2 (85.8)	-	-	1 (21.45)	-	-	1 (21.45)
**I**	T2	-	-	-	1 (46.8)	-	1 (46.8)	1 (46.8)	-	2 (93.6)	2 (93.6)	1 (−46.8)	-	-	-	2 (93.6)	1 (23.4)		1 (23.4)	1 (23.4)	-	1 (23.4)
**I**	T3	1 (253.5)	1 (−254)	2 (507)	-	-	-	-	-	-	-	-	-	-	-	-	-	-	-	-	-	-
**IV**	T1	-	-	1 (120)	1 (24)	-	1 (24)	-	-	3 (72)	1 (24)	-	2 (48)	-	-	2 (48)	-	-	2 (24)	-	-	2 (24)
**IV**	T2	2 (270)	-	5 (675)	1 (27)	-	3 (81)	-	-	4 (108)	-	1 (−27)	-	2 (54)	-	2 (54)	-	-	3 (40.5)	-	-	3 (40.5)
**IV**	T3	4 (600)	-	-	2 (60)	1 (−30)	1 (30)	2 (60)	1 (−30)	1 (30)	2 (60)	-	2 (60)	2 (60)	1 (−30)	1 (30)	1 (15)	-	3 (45)	1 (15)	-	3 (45)
**V**	T1	-	2 (−41)	8 (164)	-	-	-	-	-	-	-	-	-	-	-	-	-	-	-	-	-	-
**VI**	T1	3 (216)	-	37 (2664)	1 (14.4)	2 (−28.8)	29 (417.6)	1 (14.4)	-	29 (417.6)	5 (72)	3 (−43.2)	21 (302.4)	10 (144)	3 (−43.2)	12 (172.8)	5 (36)	1 (−7.2)	27 (194.4)	3 (21.6)	-	30 (216)
**VI**	T2	3 (252)	2 (−168)	6 (504)	-	1 (−16.8)	9 (151.2)	1 (16.8)	-	9 (151.2)	1 (16.8)	2 (−33.6)	7 (117.6)	4 (67.2)	-	6 (100.8)	1 (8.4)	2 (−16.8)	8 (67.2)	1 (8.4)	2 (−16.8)	8 (67.2)
**VI**	T3	32 (3072)	3 (−288)	12 (1152)	5 (96)	2 (−38.4)	5 (96)	7 (134.4)	2 (−38.4)	5 (96)	6 (115.2)	4 (−76.8)	4 (76.8)	8 (153.6)	1 (−19.2)	-	7 (67.2)	2 (−19.2)	5 (48)	7 (67.2)	2 (−19.2)	5 (48)
**VII**	T1	-	-	-	1 (10.5)	-	2 (21)	-	1 (−10.5)	2 (21)	-	-	3 (31.5)	1 (10.5)	-	2 (21)	1 (5.25)	-	2 (10.5)	2 (10.5)	-	1 (5.25)
**VII**	T3	-	-	-	-	-	1 (14.7)	-	-	1 (14.7)	-	-	1 (14.7)	-	-	1 (14.7)	-	-	1 (7.35)		-	1 (7.35)
**VIII**	T1	7 (252)	1 (−36)	1 (36)	-	-	-	-	-	-	-	-	-	-	-	-	-	-	-	-	-	-
**Total Sum**		5130 ^l^	−935 ^m^	6414 ^n^	278.7 ^l^	−156.9 ^m^	925.2 ^n^	272.4 ^l^	−78.9 ^m^	1089.9 ^n^	424.5 ^l^	−227.4 ^m^	693.9 ^n^	489.3 ^l^	−92.4 ^m^	620.7 ^n^	155.3 ^l^	−43.2 ^m^	481.8 ^n^	146.1 ^l^	−36 ^m^	498.2 ^n^
**Positive sum ^l^**									6896.3													
**Inverse sum ^m^**									−1569.8													
**Positive and inverse sum (l+m)**									6896.3 + (−1569.8) = 5328													
**Non-significant sum ^n^**									10,723.6													
**Absolute sum (positive sum (l) + |inverse sum| (m) + non-significant sum (n))**									6896.3 + |−1569.3| + 10,723.7 = 19,189.3													
**Score (positive and inverse sum/absolute sum)**									5328 ÷ 19,189.2 = 0.2776													

^a^ Study design codes: (I) Randomized blinded trials, (II) Randomized or blinded clinical trials, (III) Non-randomized and non-blinded clinical trials, (IV) Cohort studies, (V) Case-control studies, (VI) Nested cross-sectional studies, (VII) Cross-sectional (case-no case design) studies, (VIII) Descriptive studies, ^b^ Tertiles: (T1) Tertile 1, (T2) Tertile 2, (T3) Tertile 3 (according to each study design), ^c^ Metabolic syndrome, ^d^ Blood Glucose, ^e^ High Density Lipoprotein, ^f^ Triglycerides, ^g^ Waist Circumference, ^h^ Diastolic blood pressure, ^i^ Systolic blood pressure, ^j^ concretely here: the number of articles reporting negative effect of the food item/group on MetS with case-control design and the sample size in the first tertile among case-control studies, ^k^ the number of articles multiplied by the final score of their category; for simplification purposes, rows without any data were deleted from this table.

**Table 6 nutrients-17-02314-t006:** Coefficient table of the WDI per food component/item, with a higher absolute score indicating a stronger health impact (either positive or negative).

#	Foods ^^^	Number of Articles	Positive Sum	Negative Sum	Non-Significant Sum	Total Weight	WDI Coefficient *
1	Calorie, energy	9	93.5	−452.7	954.2	1500.3	−0.2395
2	Fiber	31	1911.9	−138.9	2313.8	4364.6	0.4062
3	Whole grains	39	2549.7	−1521	6142.5	10,213.2	0.1007
4	Carbohydrates	41	2037.2	−1552.5	4698	8287.7	0.0584
5	Refined grains	33	735.9	−2017.5	3660.8	6414.2	−0.1998
6	Legumes	38	1515.3	−375.6	4143.9	6034.8	0.1888
7	Nuts and seeds	44	3090.5	−587.7	5759.9	9438	0.2652
8	Oils	25	1310.6	−803.3	3567	5680.8	0.0893
9	Hydrogenated fat	16	388.8	−1317.6	1690.8	3397.2	−0.2734
10	Soft drinks	53	190.8	−3769.2	5730.9	9690.9	−0.3693
11	Sodium	29	520.8	−1457.9	1386.6	3365.3	−0.2784
12	Coffee, tea, and water	40	2522.6	−1119	4801.1	8442.6	0.1662
13	Protein	37	1807.1	−1216.7	4097.6	7121.3	0.0829
14	Diet drinks	8	96	−280.8	999.6	1376.4	−0.1343
15	Alcohol	21	453.6	−1026	2058	3537.6	−0.1618
16	Supplements	65	4563.3	−968.6	10,671.2	16,203	0.2219
17	Vitamins and minerals	61	6896.3	−1569.3	10,723.7	19,189.2	0.2776
18	Secondary plant metabolites	32	3938.4	−766.8	4559.1	9264.3	0.34235
19	Total fat	42	1237.2	−1590	4132.2	6959.4	−0.0507
20	Processed foods	51	717	−4829.4	7794.5	13340.9	−0.3083
21	Cholesterol, SFAs, and trans fat	27	484.2	−811.2	2965.5	4261	−0.0767
22	PUFA and MUFA	47	3729.5	−743.4	7943.1	12416	0.2405
23	Sugar	43	728.3	−2044.5	4616.4	7389.2	−0.1782
24	Fruits	67	3591.8	−676.7	7435.8	11704.2	0.2491
25	Vegetables	53	4073.4	−1009.7	7326.5	12409.5	0.2469
26	Processed meat	19	286.2	−889.2	1287.9	2463.3	−0.2448
27	Red meat	34	511.8	−2338.2	3333.6	6183.6	−0.2954
28	White meat	16	595.2	−340.8	1046.7	1982.7	0.1283
29	Fish	33	1831.2	−439.8	3300.5	5571.5	0.2497
30	Dairy	46	3609.5	−1244.4	6510.2	11364	0.2081

^^^ Refer to the Appendix A for the complete list of foods in each food group. * Either being positively associated (negative coefficient) or inversely related (positive coefficient) to the health outcome. A more positive overall score shows less adherence to the WD. MUFA, mono-unsaturated fatty acids; PUFA, poly-unsaturated fatty acids, SFAs, saturated fatty acids.

## Data Availability

The data that support the findings of this study are available from the corresponding authors upon request, as the original raw data is complex and not intuitively useful for other researchers.

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
