# Peer review of "Designing and Developing a Population/Literature-Based Westernized Diet Index (WDI) and Its Relevance for Cardiometabolic Health"

_nutrients, 2025, doi:10.3390/nu17142314_

Round 1
Reviewer 1 Report
Comments and Suggestions for Authors
To enhance readability, consider incorporating clear subheadings within each section (e.g., "Improving Results" and "Enhancing Discussion"). This would allow readers to easily navigate through the suggestions and focus on specific areas of interest.
Improving Results
- Expand Sample Size and Diversity: If possible, incorporating a more diverse population can strengthen the generalizability of the findings.
- Refine the Scoring Algorithm: Consider testing different weightings or sensitivity analyses to verify the stability of your index.
- Longitudinal Validation: If not already done, future studies could validate WDI with longitudinal data to confirm causality rather than correlation.
- Multivariate Analyses: Controlling for additional confounders (e.g., socioeconomic status, physical activity, genetic predispositions) could enhance the robustness of associations.
Enhancing Discussion
- Contextualize the Findings: Elaborate on how the results align with or diverge from previous research.
- Implications for Public Health: Strengthen the discussion on how WDI could guide dietary policy and interventions.
- Strengths & Limitations: Transparently address both the contributions of your index and potential limitations (e.g., biases in self-reported dietary data
.
Comparative Advantages of the WDI
- Comprehensiveness: Incorporates understudied components (e.g., SSBs, UPFs, supplements) absent in traditional indices (MDS, AHEI).
- Flexibility: Applicable to diverse dietary assessments (FFQs, 24-hour recalls) without requiring proprietary tools.
- Bidirectional Scoring: Unlike unidirectional indices (e.g., MDS), the WDI rewards healthy choices with positive scores, better reflecting diet quality gradients.
- Population Adaptability: Suitable for all age groups and designed to track dietary shifts over time.
Author Response
To enhance readability, consider incorporating clear subheadings within each section (e.g., "Improving Results" and "Enhancing Discussion"). This would allow readers to easily navigate through the suggestions and focus on specific areas of interest.
Reply: We thank the reviewer for the comment to improve the general readability of the manuscript. The suggestions have been included in the text. We have added 3 subheaders to the discussion: “Low ranking components of the WDI”, “High ranking components of the WDI”, “The WDI compared to other indices”.
Improving Results
- Expand Sample Size and Diversity: If possible, incorporating a more diverse population can strengthen the generalizability of the findings.
Reply: We agree with the reviewer, a bigger sample size and diversity will always improve the quality of the results and will produce more robust statistics. Unfortunately, adding more data to the WDI at this stage will not be possible, the reason being that the data was added in a systematic way and all the publications that were detected using the search strategy described in Table 2 were included into the study. This yielded a considerable number of publications ca 1500, and we included thus all eligible studies.
The following steps (title and abstract reading, as well as full text), which contained further selection criteria (Table 1) meant that 472 studies were retained for final analysis. We deem this number adequate. Changing the search strategy or modifying the selection criteria would imply a substantial change in the rationale for the development of WDI.
- Refine the Scoring Algorithm: Consider testing different weightings or sensitivity analyses to verify the stability of your index.
Reply: We thank the reviewer for this insightful comment as it has been indeed a subject of discussions between the authors. In fact, this paper is a tool-developing paper and not the validation. The validation of the index and the way to use the index is going to be explained in the next publication. Unfortunately, due to lack of longitudinal validation (see also next reply), it is not possible as of now to understand the robustness of the WDI, despite the fact that using valid methods and references, we are confident about the functionality. This means that as of this time, it is not possible to conduct sensitivity analyses.
We consider the reviewer’s comment to be of great importance even if it cannot be tackled at the moment. For this matter, we have added a specific section in the discussion where this matter is discussed, i.e. what should be tested and maybe modified in the WDI during a validation study, see lines 477-480.
- Longitudinal Validation: If not already done, future studies could validate WDI with longitudinal data to confirm causality rather than correlation.
Reply: Indeed we plan on doing so as a next step, and have mentioned this now in our discussion (see reply above).
We apologise that this was not stated clearly in the text as it is a more than valid point for a reader to wonder whether this has been validated. We have now improved the text so that the future steps are now clearer. The text now reads “Furthermore, the validation of the index in a real life population remains to be carried out”.
- Multivariate Analyses: Controlling for additional confounders (e.g., socioeconomic status, physical activity, genetic predispositions) could enhance the robustness of associations.
Reply: The reviewer raises a valid point. On the one hand, many confounders other than diet are known to have an effect on cardiometabolic health. On the other hand, Westernized lifestyle is composed of many more components than only of diet (i.e. reduced physical activity, altered circadian rhythms…). These were considerations that were carefully considered at the early drafts of the manuscript.
During the index development it was decided to focus on a diet-related version of the index and only, after potential validation in multiple studies, consider an extension toward a more complex index that considers other variables. The reason for this is twofold. First, the idea of the WDI is to be used mainly as a dietary index comparable to other indices in the field, such as the DII and MDS, which do not include other lifestyle variables. In addition, it should be an index that can be calculated using data typically collected in any study, with accessible validated methods such as FFQ or 24h recalls. Second, although the index has the potential to be expanded and does not contain all the variables of a potential Western lifestyle adjustment for confounders, when using the index this can be done in the same way as for other indices. This means that when conducting a multivariate analysis in a given population, there should be no limitation to adjust for additional confounders.
To enhance the manuscript and make more clear for the reader the future perspectives and compatibility with controlling for confounders, we have added these aspects discussed above into the text. (Lines 477-480)
Enhancing Discussion
- Contextualize the Findings: Elaborate on how the results align with or diverge from previous research.
Reply: We have tried to capture the comparison between the WDI and other indices in the original manuscript, but we have now overhauled the subsection of the discussion in several ways. First, a sub header “The WDI compared to other indices” has been included to highlight the content. Second, we have improved the text in lines 415-421 and 455-457, providing further comparisons and improving reader-friendliness.
- Implications for Public Health: Strengthen the discussion on how WDI could guide dietary policy and interventions.
Reply: We appreciated this insightful comment and agree that we could have discussed the implications of the index in public health more thoroughly. For this reason, we have added additional comments in lines 462-463.
- Strengths & Limitations: Transparently address both the contributions of your index and potential limitations (e.g., biases in self-reported dietary data
Reply: We had attempted to achieve this within the last two paragraphs of the discussion, just before the conclusions. We have by now further revised and improved this section.
.
Comparative Advantages of the WDI
- Comprehensiveness: Incorporates understudied components (e.g., SSBs, UPFs, supplements) absent in traditional indices (MDS, AHEI).
- Flexibility: Applicable to diverse dietary assessments (FFQs, 24-hour recalls) without requiring proprietary tools.
- Bidirectional Scoring: Unlike unidirectional indices (e.g., MDS), the WDI rewards healthy choices with positive scores, better reflecting diet quality gradients.
- Population Adaptability: Suitable for all age groups and designed to track dietary shifts over time.
Reply: We appreciate the reviewers’ recognition of the unique strengths of the WDI. Indeed, its comprehensiveness—especially the inclusion of often-overlooked dietary components such as SSBs, UPFs, and supplements—sets it apart from conventional indices. We also agree that the flexibility of the WDI, allowing it to be used across multiple dietary assessment methods without proprietary tools, enhances its practical utility in diverse research settings. The bidirectional scoring approach was deliberately chosen to more accurately capture the full spectrum of dietary quality. Finally, the adaptability of the WDI across age groups and its capacity to monitor temporal dietary changes were key priorities in its development. We thank the reviewers for highlighting these comparative advantages.

Reviewer 2 Report
Comments and Suggestions for Authors
Dear Authors,
Diet quality indexes are a very important tool for nutritional epidemiology and for identifying correlations between dietary habits and health outcomes. With the evolution of eating habits, it is essential to manage instruments that can capture these changes and identify their impact.
The extensive work presented in this article certainly represents an important step in this direction and could provide a new tool for future research. However, I believe that the material currently submitted to the journal Nutrients requires extensive revision.
Material and Methods: attaching two tables and one figure to describe the bibliographic research methodology is redundant (also lines 160-168), and the text is also too detailed. It should be shortened to highlight the subsequent parts.
Line 138: which WD components have been considered?
The scoring algorithm is very complex, and it is not easy to understand the logical process that led to its development, and, above all, why successive multiplication and division operations were performed (lines 225-235). Are there any bibliographical references that can be consulted to better understand and, above all, how you validated this scoring method? This is a key part of the article and is difficult to understand.
The allocation in tertiles (T1, T2 and T3) does not provide any indication about the number (or ranges) of each tertile.
The tables with examples of score calculations should also include real examples. For example, could Table 7 (on page 13, not Table 5) be used to calculate the overall score for a food diary?
Table 6 is also difficult to read.
It may be interesting to use WDI on samples already analysed with other quality indices to verify its reliability.
Overall, this is an interesting study because, in addition to several diet indexes previously evaluated, it assesses the level of processing and includes both diet drinks and soft drinks, but it is very complex and both the validation methods and the scientific basis underlying the algorithm are unclear.
Discussion: could be more concise.
Sentence line 448: based on what data could it be said that WDI is suitable for such a wide range of age groups?
Minor:
Supplementary table 2 (line 419) is not available and also the link at line 480 is not working.
References are not in line with Nutrients editorial style.
Author Response
Diet quality indexes are a very important tool for nutritional epidemiology and for identifying correlations between dietary habits and health outcomes. With the evolution of eating habits, it is essential to manage instruments that can capture these changes and identify their impact.
The extensive work presented in this article certainly represents an important step in this direction and could provide a new tool for future research. However, I believe that the material currently submitted to the journal Nutrients requires extensive revision.
Reply: We thank the reviewer for sharing his frank judgement and the importance of dietary indices in nutritional epidemiology and for the appraisal to the WDI. We have carefully considered the comments and tried to improve the manuscript so it can comply better with the standards of the field.
- Material and Methods: attaching two tables and one figure to describe the bibliographic research methodology is redundant (also lines 160-168), and the text is also too detailed. It should be shortened to highlight the subsequent parts.
Reply: We agree that having two tables and one figure to describe the bibliographic research is perhaps too much and partially draws the attention from the more interesting subject of the index development. We have now moved table 2 to the supplementary document and kept only the figure and a short explanation of the methodology in the main body of the manuscript.
- Line 138: which WD components have been considered?
Reply: The WD components are found in the former table 2 line #2. We appreciate the comment from the reviewer as it is clear now that this presentation of the data is not optimal to the reader and has now been improved. The complete list is now found in a supplementary table first introduced at line 152.
- The scoring algorithm is very complex, and it is not easy to understand the logical process that led to its development, and, above all, why successive multiplication and division operations were performed (lines 225-235). Are there any bibliographical references that can be consulted to better understand and, above all, how you validated this scoring method? This is a key part of the article and is difficult to understand.
Reply: The complexity of the scoring index was one of our main concerns. This is the reason why such an extensive amount of tables is shown throughout the document. We have re-checked the entire text and overhauled the parts that are relevant to understanding how the index was developed.
Of note, the methodology for the scoring algorithm is based on that of the dietary inflammatory index (DII), a validated methodology, which is referenced throughout the text:
doi:10.1017/S1368980013002115; https://doi.org/10.1093/advances/nmy071
Also, The levels of evidence and their role in evidence-based medicine - PubMed was used as one of the references for developing the index, which is cited in the publication now (line 203).
- The allocation in tertiles (T1, T2 and T3) does not provide any indication about the number (or ranges) of each tertile.
Reply: We thank the reviewer for highlighting the need for clarity regarding the tertile distributions. We have now included a detailed table with tertile ranges and sample sizes for each study design to foster transparency. However, we chose not to include this level of detail in the main manuscript or supplementary materials to maintain focus and readability. These data are available upon request, as part of our data-sharing policy, which is why we did not include more details on the breakdown of tertiles in the manuscript itself. We have added a sentence as an example to further clarify the tertiles in one of the study designs in lines 208-210.
|
Range of Tertiles |
Tertile 1 |
Tertile 2 |
Tertile 3 |
|
Clinical trials |
Min = 12, Max = 43 |
Min = 44, Max = 84 |
Min = 87, Max = 5220 |
|
Cohort |
Min = 32, Max = 1915 |
Min = 2045, Max = 5373 |
Min = 5509, Max = 14618 |
|
Case-control |
Min = 258, Max = 442 |
|
Min = 1872, Max = 2000 |
|
Cross-sectional |
Min = 76, Max = 200 |
Min = 504, Max = 1004 |
Min = 1094, Max = 13170 |
|
Nested |
Min = 73, Max = 1270 |
Min = 1301, Max = 6308 |
Min = 6367, Max = 161326 |
|
Descriptive designs |
Min = 189, Max = 290 |
Min = 320, Max = 637 |
Min = 1040, Max = 25506 |
- The tables with examples of score calculations should also include real examples. For example, could Table 7 (on page 13, not Table 5) be used to calculate the overall score for a food diary?
Reply: Table 7 shows the final coefficient of each food group in WDI. Therefore, it will be used in the next step to calculate the final score for each individual. It means that the last column “WDI coefficient” is the final result following the steps explained before for each food group. To clarify the usage of this table, the following sentence was added to the manuscript “The calculated coefficients will be later used to validate the index and develop a calculations algorithm.”
Also, explanations added to supplementary table 2. In the future when we use real world data to validate the index, we will be able to calculate it according to food diary or FFQ.
- Table 6 is also difficult to read.
Reply: Indeed, fitting big tables in the manuscript is not easy considering the limitations we have regarding typesetting, and we leave it for the journal’s layout team to make it clearer and fit to the template. Nevertheless, we have tried to make some changes to improve reader-friendliness. Moreover, this table is only an example to better understand the text, there are clean versions of this table for each group coefficient calculation of the full dataset, available upon request.
- It may be interesting to use WDI on samples already analysed with other quality indices to verify its reliability.
Reply: Indeed a major limitation of the current study is that so far a validation either with already analysed samples or with a combination of validation + test cohorts and comparisons with other indices is missing. The main reason for not doing so is that we believe the current manuscript being already very complex and that adding a validation step would hamper the integrity of the detailed methodology described, which is deemed needed to fully understand the index.
We have addressed this issue at the end of the discussion in lines 494-498, which reads now:
“Furthermore, the main limitation of the index is a validation in a real life population remains to be done. Without this step, it is not possible at the moment to know the robust-ness of the index and its association with cardiometabolic diseases. It will therefore re-main a challenge for the validation step to confirm the test the sensitivity of the index and whether other weighing approaches may impact its power as a public health tool.”
- Overall, this is an interesting study because, in addition to several diet indexes previously evaluated, it assesses the level of processing and includes both diet drinks and soft drinks, but it is very complex and both the validation methods and the scientific basis underlying the algorithm are unclear.
Reply:
We thank the reviewer for this supportive comment. As mentioned before, we are well aware of the complexity of the topic and we have used this revision to improve the explanations on the methodology and the rationale behind. We have further made it clear that the methodology is based on that used for developing the DII, which should reassure the readers that it is a validated method, even if in this particular case it has not yet been further validated.
Regarding formal validation of the index we agree that this is a key step that has to be carried out before the index can be fully used in public health and/or research. It is our goal to perform this validation in a next step. We believe that due to the amount of data concentrated in the current manuscript it is not feasible to include the validation at this stage, as this would equally produce much data and would overload the reader. We have added a few lines at the end of the discussion addressing this major limitation.
- Discussion: could be more concise.
Reply: We have further revised the Discussion section to improve clarity and conciseness by removing redundant statements and streamlining key points. We believe these changes enhanced the readability and focus of the discussion while retaining the necessary depth of interpretation.
- Sentence line 448: based on what data could it be said that WDI is suitable for such a wide range of age groups?
Reply: With this sentence we meant to indicate that we expect the index to work on multiple age groups because all these groups were represented in significant numbers in the articles used during data extraction step. We have rephrased the text to read:
“The WDI was designed using a wide range of data comprising populations from all age groups and thus it is expected to be applicable across those same groups, including children and adolescents as well as adults and older people.”
This claim can also be supported by the supplementary figure 2, showing the age range that the reviewed articles covered.
Furthermore, it is clear that the main limitation of the manuscript is the validation of the index on real-life populations. We have addressed this in the text as a major limitation and intend to validate the index in the following steps.
Minor:
- Supplementary table 2 (line 419) is not available and also the link at line 480 is not working.
Reply: We apologize, indeed supplementary material was not uploaded properly. This has now been fixed. As for the mentioned line, it is part of the template, we believe that this can only be modified by the editor upon article acceptance.
- References are not in line with Nutrients editorial style.
Reply: We have updated the references to match the official MDPI template.

Reviewer 3 Report
Comments and Suggestions for Authors
The manuscript is well-written, and the methodology and results could be improved. However, the article's biggest problem is that it doesn't contribute new knowledge. All the results are already known, so I don't think it's publishable
Author Response
The manuscript is well-written, and the methodology and results could be improved. However, the article's biggest problem is that it doesn't contribute new knowledge. All the results are already known, so I don't think it's publishable.
Reply: We thank the reviewer for taking the time to read the manuscript. We have revised the manuscript in response to the reviewers’ and editor’s comments to the best of our ability, and we believe the manuscript provides very novel insights and, more importantly, presents a valuable and novel nutritional tool for public health applications. Below, we outline several aspects considering the novelty and merits of the manuscript:
- The primary objective of the WDI is not to redefine the components of a Western diet or their health effects—these are indeed well established and widely supported by existing research, with our findings aligning with the literature. Rather, the developed WDI is intended as a practical tool for dietary assessment.
- Tools for dietary assessment, such as the Mediterranean Diet Score (MDS) and the Dietary Inflammatory Index (DII), are designed to quantify adherence to specific dietary patterns known to influence human health. This approach is essential because, although individual diets may appear similar, they can differ significantly in composition. In public health and epidemiology, there is a critical need for standardized methods to compare dietary patterns across individuals and populations on a holistic scale, as well as to inform dietary recommendations. Such comparisons would not be possible without indices such as the MDS, DII, or the WDI.
- The manuscript, and consequently the WDI, aim to provide researchers with a comprehensive tool to categorize individuals’ diets. Its methodology and rationale align with those of many established dietary indices.
- Unlike most indices, which focus either on adherence to beneficial dietary patterns or on minimizing negative aspects of unhealthy diets (such as oxidative stress, inflammation, and hypertension), the WDI is novel in that it evaluates dietary intake in a bidirectional manner.
- The WDI incorporates aspects often overlooked by traditional indices, such as the consumption of sugar-sweetened beverages and ultra-processed foods, into its dietary score calculation.
We would be more than happy to address any additional concerns the reviewer may have.

Round 2
Reviewer 2 Report
Comments and Suggestions for Authors
Dear Authors,
Thank you for the correzione done.
Reviewer 3 Report
Comments and Suggestions for Authors
The authors improved the manuscript's form, but it did not change its substance. Consequently, it does not contribute any new knowledge.